# Association between Dietary Calcium and Potassium and Diabetic Retinopathy: A Cross-Sectional Retrospective Study

**DOI:** 10.3390/nu14051086

**Published:** 2022-03-04

**Authors:** Yuan-Yuei Chen, Ying-Jen Chen

**Affiliations:** 1Department of Pathology, National Defense Medical Center, Tri-Service General Hospital Songshan Branch and School of Medicine, Taipei 114, Taiwan; fu84fu840618@gmail.com; 2Department of Pathology, National Defense Medical Center, Tri-Service General Hospital and School of Medicine, Taipei 114, Taiwan; 3Department of Ophthalmology, National Defense Medical Center, Tri-Service General Hospital and School of Medicine, Taipei 114, Taiwan

**Keywords:** micronutrients, dietary calcium, retinopathy

## Abstract

Background: Micronutrients are considered to have an important role in metabolic process. The relationships between micronutrients and diabetic complication, such as retinopathy, are rarely discussed. The main purpose of the current study was to investigate the relationship between dietary micronutrients and diabetic retinopathy in an adult population. Methods: 5321 participants from National Health and Nutritional Examination Survey (NHANES) 2005–2008 were included in this cross-sectional study. Diabetic retinopathy was diagnosed by the severity scale of the Early Treatment Diabetic Retinopathy Study (ETDRS) using nonmydriatic fundus photography. Micronutrients were assessed by 24-h dietary recall. The relationship between dietary micronutrients and the occurrence of diabetic retinopathy was analyzed by a logistic regression model. Results: Dietary calcium and potassium were inversely associated with diabetic retinopathy (OR: 0.729, 95%CI: 0.562–0.945; OR: 0.875, 95%CI: 0.787–0.973). Higher quartile of dietary calcium and potassium was associated with lower occurrence of diabetic retinopathy (OR: 0.664, 95%CI: 0.472–0.933; OR: 0.700, 95%CI: 0.495–0.989). Furthermore, increased amounts of dietary calcium and potassium were significantly associated with reduced occurrence of diabetic retinopathy (OR: 0.701, 95%CI: 0.546–0.900; OR: 0.761, 95%CI: 0.596–0.972). Conclusions: Higher levels of dietary calcium and potassium are suggested to reduce the risk of diabetic retinopathy with dose–response effect. The evaluation of dietary micronutrients might be a part of treatment for patients with diabetic complications.

## 1. Introduction

Micronutrients are groups of essential dietary elements, including vitamins and trace minerals, required by the human body in varying quantities to regulate physiological processes to maintain body functions and health [1]. Micronutrients play an important role in the regenerative process, immune support, and antioxidant development [2,3]. The adequate requirement for essential dietary micronutrients can minimize risks of nutrient excess or deficiency [4]. Impaired homeostasis of micronutrients may contribute to the development of diseases and pathologic states [5]. Increased levels of dietary phosphorus are suggested to cause renal failure, cardiovascular disease, and osteoporosis [6,7]. Dietary magnesium is inversely related to several chronic systemic diseases, such as hypertension, ischemic heart disease, and stroke [8]. Dietary calcium has been suggested a possible risk factor for metabolic syndrome [9].

Diabetic retinopathy is a common complication of diabetes that is suggested to be the leading cause of visual impairment and visual loss in adults [10,11,12]. In an epidemiologic study for the adult population in the United States, there were about 30 to 40% of diabetic patients at risk of retinopathy [13]. Diabetic retinopathy is one of the comorbidities for many other diabetes-related disorders, such as osteoporosis, peripheral neuropathy, chronic kidney disease, and cardiovascular events, which reduce the life quality and lead to increased mortality [14,15,16]. Poor glycemic control, longer diabetes duration, and higher glycosylated hemoglobin are reported to be independent risk factors of diabetic retinopathy [17].

After reviewing the database of the National Health and Nutrition Examination Survey (NHANES), the relationship between dietary trace elements and type 2 diabetes has been proposed that higher dietary calcium is related to decreased risk of type 2 diabetes [18]. However, the potential effect of dietary micronutrients on diabetic retinopathy has not been reported yet. Therefore, the main purpose of the current study was to investigate the relationship between dietary micronutrients and diabetic retinopathy among a nationally representative general adult population in the United States.

## 2. Materials and Methods

### 2.1. Study Design and Participant Recruitment

All study information was collected from the National Health and Nutrition Examination Survey (NHANES) 2005–2008. NHANES is a major program of the National Centers for Health Statistics (NCHS). This nationally representative survey engages a variety of population health and nutrition measurements in the United States. A total of 5321 eligible participants who had received retinal imaging examination were enrolled in this study. According to the flow chart shown in Figure 1, we excluded those who had (1) missing data regarding dietary intake of trace elements, energy, carbohydrate, and sugar (*n* = 96); (2) missing data regarding their serum sample for laboratory examination (*n* = 6); and (3) missing data from the questionnaire for race/ethnicity and smoking history (*n* = 8). Ethics approval was accepted by the institutional review board of the NCHS and study design was confirmed in accordance with the Helsinki Declaration. All articipants provided informed consent before enrollment.

### 2.2. Diagnosis of Diabetic Retinopathy

According to the severity scale provided by the Early Treatment for Diabetic Retinopathy Study (ETDRS), retinopathy is characterized by hard exudates, vitreous hemorrhage, cotton-wool spots, and intra-retinal microvascular changes over retinal tissue [19]. A non-mydriatic fundus photography (TRC-NW6S; Topcon, Tokyo, Japan) was used for detecting diabetic retinopathy in the survey based on the NHANES Digital Grading Protocol.

### 2.3. Assessment of Dietary Trace Elements

Twenty-four-hour dietary recall was used to estimate the dietary intakes of micronutrients, including calcium, phosphorous, magnesium, zinc, copper, sodium, potassium, and selenium, by the multi-pass approach [20]. This respondent-driven method is a brief and precise list for collecting foods and beverages digested by an individual during 24 h. Detailed descriptions of dietary interview methods were available from dietary interviewers’ procedure manuals in NHANES.

### 2.4. Relevant Variables

Characteristic data included age, gender, races/ethnicity, and cigarette smoking status. Race/ethnicity was divided into 5 groups, including Hispanic, non-Hispanic White, non-Hispanic Black, non-Hispanic Asian, and other. Cigarette smoking status was assessed by asking participants the question “do you now smoke cigarettes?”. Serum laboratory data such as glucose, alanine aminotransferase, and hemoglobin were measured by standard procedures.

### 2.5. Statistical Analysis

All statistical analyses were performed using the Statistical Package for the Social Sciences, version 22.0 (SPSS Inc., Chicago, IL, USA). Significant differences between participants with and without retinopathy were estimated by chi-square test. A *p* value less than 0.05 was statistically significant. Association between dietary micronutrients and odds ratio (OR) for diabetic retinopathy was performed by a multivariable logistic regression model, which was adjusted for age, gender, race/ethnicity, serum laboratory data, and cigarette smoking status. Optimal cutoff points of dietary micronutrients for predicting the occurrence of diabetic retinopathy were conducted by a receiving operating characteristic (ROC) curve analysis.

## 3. Results

### 3.1. Study Population Characteristics

Table 1 listed characteristic information of 5211 eligible participants with (*n* = 696) and without diabetic retinopathy (*n* = 4515). The mean age of those with and without retinopathy is 62.43 ± 11.79) and 58.96 ± 12.42 years. People with diabetic retinopathy have significantly lower dietary trace elements, serum glucose, hemoglobin, dietary energy, carbohydrate, and sugar (*p* < 0.05).

### 3.2. Association between Dietary Trace Elements and Diabetic Retinopathy

In Figure 2, dietary micronutrients, such as calcium, phosphorous, magnesium, and potassium had significant association with the presence of diabetic retinopathy (OR: 0.698, 95% corresponding interval (CI): 0.549–0.889; OR: 0.796, 95%CI: 0.657–0.965; OR: 0.364, 95%CI: 0.156–0.848; OR: 0.875, 95%CI: 0.795–0.964) (*p* < 0.001), respectively. After fully adjusting, relevant variables included age, gender, race/ethnicity, serum glucose, ALT, hemoglobin, and smoking history; only dietary calcium and potassium remained significant with the presence of diabetic retinopathy (OR: 0.729, 95%CI: 0.562–0.945; OR:0.875, 95%CI: 0.787–0.973), respectively.

### 3.3. Association between Quartile of Dietary Calcium, Potassium, and Diabetic Retinopathy

After categorizing dietary calcium into quartiles (Table 2), we found that quartile 2 and quartile 3 had inverse association with diabetic retinopathy (OR: 0.622, 95%CI: 0.444–0.871; OR: 0.664, 95%CI: 0.472–0.933). Quartile 3 of potassium was also inversely associated with the presence of diabetic retinopathy (OR: 0.700, 95%CI: 0.495–0.989).

### 3.4. Association between Dietary Calcium and the Risk of Diabetic Retinopathy

To determine the ability of dietary calcium and potassium to predict the risk of diabetic retinopathy, the cutoff points for dietary calcium and potassium were estimated by ROC analysis (Figure 3). Areas under ROC of dietary calcium and potassium were 0.540 (95%CI: 0.517–0.563), 0.545 (0.521–0.568), and cutoff point was 550.501 (sensitivity: 69.9%; specificity: 38.1%), 2262.500 (sensitivity: 57.2%; specificity 50.9%), respectively.

In Table 3, dietary calcium and potassium were significantly correlated with reduced occurrence of diabetic retinopathy (OR: 0.701, 95%CI: 0.546–0.900; OR: 0.761, 95%CI: 0.596–0.972) after fully adjusting for relevant variables.

## 4. Discussion

In the present study, we found that dietary calcium and potassium are significantly associated with the presence of diabetic retinopathy. Participants with higher levels of dietary calcium and potassium have less occurrence of retinopathy. As far as we know, the current study was the first to investigate the relationship between dietary micronutrients and diabetic retinopathy in a cross-sectional study composed of a United States adult population.

Growing evidence have proposed that micronutrients are pivotal and have an important role in various metabolic processes [21]. In a population-based study, serum phosphorus is suggested as a risk factor of metabolic syndrome in a Taiwanese elderly population [22]. Zhang et al. and colleagues proposed that high serum trace elements levels, including copper, zinc, and magnesium, are associated with hypertension in a Chinese population. Numerous studies have demonstrated that high concentrations of calcium are associated with increased risk of type 2 diabetes [23,24]. Calcium is suggested to contribute to impaired glucose metabolism and insulin resistance [25]. A recent study has demonstrated that higher serum calcium level may be a risk factor for retinopathy in people with diabetes [26]. In our study, we found an opposite finding that dietary calcium is associated with reduced risk of retinopathy. This discrepancy may be due to differences in the underlying mechanisms between intracellular and oral intake of calcium.

Pittas et al. reported that combined oral vitamin D and calcium supplementation may have a role in the prevention of diabetes [27]. In a cross-sectional study assessing data from NHANES, low dietary calcium is related to increased risk of diabetes in an adult population [18]. Kim et al. proposed that serum calcium is not associated with diabetes development, while higher dietary calcium is associated with a decreased risk of diabetes [28]. There are some possible pathways suggested to explain this relationship. An experimental study showed that dietary calcium intake could increase the extracellular calcium, which affects the beta cells of pancreas and improve insulin secretion and insulin resistance [29]. Gomes et al. demonstrated that lower dietary calcium might increase the risk of diabetes by indirectly causing changes in gastrointestinal hormones or the intestinal microbiome and integrity [30]. In addition, high calcium intake is suggested to be correlated with a decreased risk of incident atherosclerosis [31]. There is evidence suggesting that higher calcium intake is associated with better lipid profiles and lower blood pressure [32,33], which may be beneficial for cardiovascular health [34]. According to the above findings, we speculated that high dietary calcium might have an important role in improving microvascular change in patients with diabetic retinopathy.

The relationship between potassium and risk of diabetic retinopathy has been reported in previous studies. In two studies, neither dietary potassium nor sodium was associated with risk of diabetic retinopathy in Japanese and European participants with diabetes [35,36]. In contrast, Chatterjee et al. has reported that serum potassium was associated with impaired insulin sensitivity and increased risk of diabetic retinopathy [37,38]. In a systemic review, low serum potassium was considered a risk of incident diabetes and more consumption of potassium-rich foods was associated with lower risks of diabetes [39]. In addition, potassium was suggested to have an anti-inflammatory function by suppressing T-cell proliferation and inducing Foxp3+ cells expression [40]. Evidence supports the view point that dietary potassium has a protective effect on the endothelium, which may prevent vascular injury against oxidative stress [41,42].

Some limitations of this study should be noted. First, even though the relationship between dietary micronutrients and diabetic retinopathy were clarified, the causal relationship could not be determined because of the cross-sectional design in this study. Second, institutionalized citizens were not included in the NHANES database; therefore, the prevalence of diabetic retinopathy might be underestimated. Next, data about dietary supplements had missing values, which was why we did not collect supplementary data. Nonetheless, dietary intake was collected by the recall. Lastly, the findings of the current study might not be universal because the survey was only held in a specific geographic region. Further research is necessary to produce a more definitive and general recommendation for nutrition.

## 5. Conclusions

In the current study we provided strong evidence that dietary calcium and potassium were significantly associated with diabetic retinopathy in an adult population of the United States. Higher levels of dietary calcium and potassium may predict reduced occurrence of diabetic retinopathy with dose–response effect. Our results addressed that the evaluation of dietary micronutrients might be a part of follow-up visits in people with diabetes. Further studies are necessary to recognize the potential mechanism in altered calcium and potassium homeostasis in the pathogenesis of diabetic complications.

## Figures and Tables

**Figure 1 nutrients-14-01086-f001:**
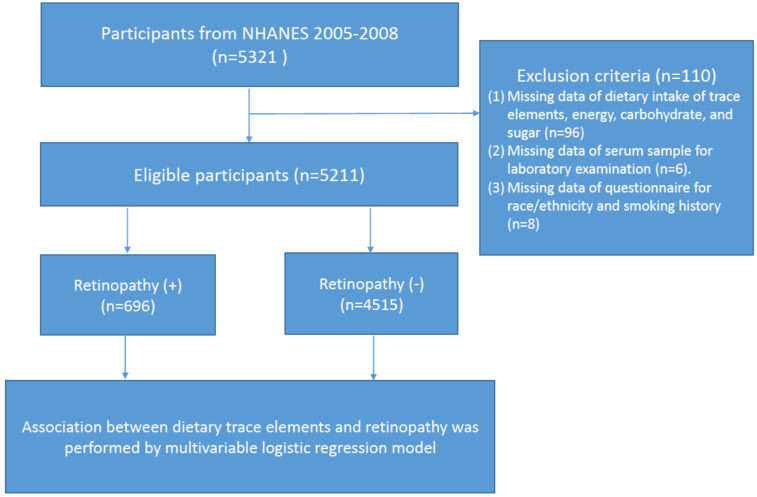
Flow chart of the study.

**Figure 2 nutrients-14-01086-f002:**
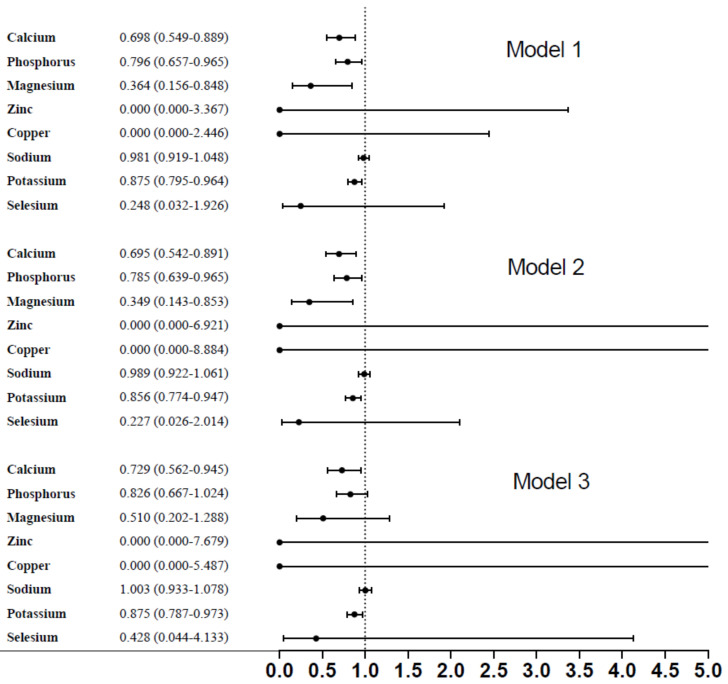
Association between dietary micronutrients and diabetic retinopathy.

**Figure 3 nutrients-14-01086-f003:**
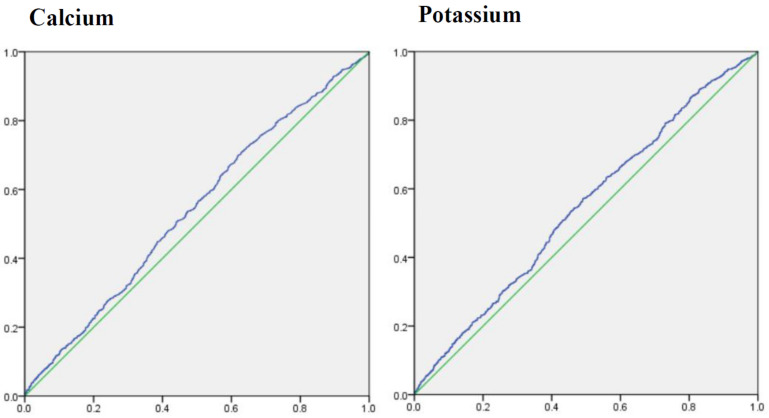
Areas under receiver operating characteristic curve for calcium and potassium.

**Table 1 nutrients-14-01086-t001:** Characteristics of Study Population.

Variables	Diabetic Retinopathy (*n* = 696)	No Diabetic Retinopathy (*n* = 4515)	*p*-Value
Continuous variables, mean (SD)			
Age (years)	62.430 (11.790)	58.961 (12.421)	<0.001
Serum glucose (mg/dL)	135.830 (71.782)	102.671 (31.910)	<0.001
Alanine aminotransferase (U/L)	26.191 (34.730)	25.530 (16.942)	0.428
Hemoglobin (g/dL)	14.071 (1.682)	14.290 (1.522)	<0.001
Dietary calcium (g)	0.796 (0.482)	0.866 (0.531)	<0.001
Dietary phosphorus (g)	1.164 (0.548)	1.259 (0.623)	<0.001
Dietary magnesium (g)	0.266 (0.139)	0.287 (0.142)	<0.001
Dietary zinc (g)	0.011 (0.006)	0.012 (0.011)	0.027
Dietary copper (g)	0.001 (0.001)	0.001 (0.001)	0.030
Dietary sodium (g)	3.003 (1.552)	3.197 (1.711)	0.005
Dietary potassium (g)	2.437 (1.135)	2.628 (1.224)	<0.001
Dietary selenium (mg)	0.098 (0.051)	0.103 (0.058)	0.038
Dietary energy (kcal)	1848.71 (839.15)	2010.08 (922.90)	<0.001
Dietary carbohydrate (gm)	221.47 (111.48)	242.64 (113.72)	<0.001
Dietary sugar (gm)	96.63 (74.96)	110.60 (70.55)	<0.001
Category variables, (%)			
Gender (male)	389 (55.9)	2257 (50.0)	0.004
Non-Hispanic white	298 (42.8)	2560 (56.7)	0.400
Cigarette smoking status	120 (17.2)	817 (18.1)	0.603

**Table 2 nutrients-14-01086-t002:** Association between quartiles of calcium and potassium and the presence of diabetic retinopathy.

		Model 1 OR (95% CI)	*p* Value	Model 2 OR (95% CI)	*p* Value	Model 3 OR (95% CI)	*p* Value
	*Retinopathy*						
Calcium	Q1 vs. Q4	0.802 (0.594–1.084)	0.152	0.790 (0.584–1.070)	0.128	0.814 (0.592–1.120)	0.206
	Q2 vs. Q4	0.618 (0.450–0.850)	0.003	0.606 (0.440–0.835)	0.002	0.622 (0.444–0.871)	0.006
	Q3 vs. Q4	0.631 (0.459–0.869)	0.005	0.628 (0.454–0.870)	0.005	0.664 (0.472–0.933)	0.018
Potassium	Q1 vs. Q4	0.899 (0.657–1.229)	0.504	0.846 (0.616–1.161)	0.300	0.842 (0.604–1.173)	0.309
	Q2 vs. Q4	0.816 (0.593–1.123)	0.212	0.766 (0.553–1.059)	0.107	0.778 (0.555–1.092)	0.147
	Q3 vs. Q4	0.714 (0.518–0.983)	0.039	0.663 (0.476–0.924)	0.015	0.700 (0.495–0.989)	0.043

Adjusted variables: Model 1: unadjusted; Model 2: age, gender, race/ethnicity; Model 3: age, gender, race/ethnicity, serum glucose, ALT, hemoglobin, cigarette smoking status.

**Table 3 nutrients-14-01086-t003:** Association between cutoff points of calcium and potassium, and the presence of diabetic retinopathy.

Cutoff Points	Model 1 OR (95% CI)	*p* Value	Model 2 OR (95% CI)	*p* Value	Model 3 OR (95% CI)	*p* Value
	*Retinopathy*					
Calcium 550.501	0.683 (0.540–0.864)	<0.001	0.677 (0.534–0.858)	<0.001	0.701 (0.546–0.900)	0.005
Potassium 2262.50	0.754 (0.601–0.946)	0.015	0.724 (0.574–0.914)	0.007	0.761 (0.596–0.972)	0.029

Adjusted covariates: Model 1: unadjusted; Model 2: age, gender, race/ethnicity; Model 3: age, gender, race/ethnicity, serum glucose, ALT, hemoglobin, cigarette smoking status.

## Data Availability

Data available in a publicly accessible repository.

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
