# Peer review of "Association between Dietary Calcium and Potassium and Diabetic Retinopathy: A Cross-Sectional Retrospective Study"

_nutrients, 2022, doi:10.3390/nu14051086_

Round 1
Reviewer 1 Report
The manuscript entitled “Association between dietary micronutrients and retinopathy” is simple, but straightforward and relatively short. By using NHANES data, the authors conclude that Ca and P are important minerals for retinopathy. However, the authors did not fully expect the potential mechanism of how the lack of Ca and P may cause retinopathy. Retinopathy is closely associated with ocular inflammation and ER stress; therefore, authors may consider writing how Ca and P prevent pathological consequences of retinopathy (i.e. suppression of inflammatory responses and ER stress).
Besides, please consider to correct as follows:
[L107] Carbohydrate → carbohydrate
[L133] delete space before ‘ROC’
Author Response
March 1, 2022
Dear editor:
Thank you for your encouraging letter concerning our manuscript entitled “Association between dietary micronutrients and retinopathy” by Yuan-Yuei Chen et al, for publication in Article of Nutrients.
We are extremely grateful to you and the reviewers for the constructive critique of our manuscript. We have responded to each of the comments of the referees on separate sheets and deeply appreciated your suggestions that have led to a significant improvement in this article. In response to your comments, we have revised the manuscript to enhance article readability. Several new sections of text and tables are added. We have also reedited the abstract, introduction, results, methods, and discussion sections. All the changes are labeled in red color. This manuscript has been edited by the American Journal Experts. Hope our revisions have fully addressed your concerns. Accordingly, we resubmit this article to Nutrients.
We look forward to your prompt reply.
Yours sincerely,
Wei-Liang Chen, M.D. Ph.D.
Division of Geriatric Medicine, Department of Family Medicine, Tri-Service General Hospital, National Defense Medical Center,
Number 325, Section 2, Chang-gong Rd, Nei-Hu District, 114, Taipei, Taiwan, ROC.
Tel: +886-2-87923311 ext. 16567
Fax: +886-2-87927057
E-mail: weiliang0508@gmail.com
Answer to Reviewer's comments
Thank you for your positive comments on this manuscript. The responses to the raised questions are below.
Reviewer 1
General comment:
- The manuscript entitled “Association between dietary micronutrients and retinopathy” is simple, but straightforward and relatively short.
Response: Thank you for your thorough review and salient observations. We have revised the title based on your recommendation.
Association between dietary calcium and potassium and retinopathy: a cross-sectional retrospective study
- By using NHANES data, the authors conclude that Ca and P are important minerals for retinopathy. However, the authors did not fully expect the potential mechanism of how the lack of Ca and P may cause retinopathy. Retinopathy is closely associated with ocular inflammation and ER stress; therefore, authors may consider writing how Ca and P prevent pathological consequences of retinopathy (i.e. suppression of inflammatory responses and ER stress).
Response: Thank you for your thorough review and salient observations. We have added the potential mechanism into the manuscript based on your recommendation.
In addition, potassium was suggested to have an anti-inflammatory function by suppressing T-cell proliferation and inducing Foxp3+ cells expression1. Several evidence supported this view point that dietary potassium had protective effect on endothelium, which may prevent vascular injury against oxidative stress2,3.
Reference
- Khalili, H.; Malik, S.; Ananthakrishnan, A.N.; Garber, J.J.; Higuchi, L.M.; Joshi, A.;
Peloquin, J.; Richter, J.M.; Stewart, K.O.; Curhan, G.C., et al. Identification and Characterization of a Novel Association between Dietary Potassium and Risk of Crohn's Disease and Ulcerative Colitis. Frontiers in immunology 2016, 7, 554, doi:10.3389/fimmu.2016.00554.
- Kido, M.; Ando, K.; Onozato, M.L.; Tojo, A.; Yoshikawa, M.; Ogita, T.; Fujita, T.
Protective Effect of Dietary Potassium Against Vascular Injury in Salt-Sensitive Hypertension. 2008, 51, 225-231.
- Smiljanec, K.; Mbakwe, A.; Ramos Gonzalez, M.; Farquhar, W.B.; Lennon, S.L.
Dietary Potassium Attenuates the Effects of Dietary Sodium on Vascular Function in Salt-Resistant Adults. 2020, 12, 1206.
- Besides, please consider to correct as follows:
[L107] Carbohydrate → carbohydrate
[L133] delete space before ‘ROC’
Response: Thank you for your thorough review and salient observations. We have revised the sentence based on your recommendation.
People with diabetic retinopathy has significantly lower dietary trace elements, serum glucose, hemoglobin, dietary energy, carbohydrate, and sugar (p < 0.05).
To determine the ability of dietary calcium and potassium to predict the risk of diabetic retinopathy, the cutoff points for dietary calcium and potassium were estimated by ROC analysis (Figure 3).
Last, we are deeply honored by the time and effort you spent in reviewing this manuscript. In reviewing and revising our text, we are motivated to read more and thus learn more from your criticisms.

Reviewer 2 Report
I’ve read with attention the paper of Chen and Chen that is potentially of interest. The background and aim of the study have been clearly defined. The methodology applied is overall correct, the results are reliable and adequately discussed. However, the authors should make the title more informative: it should be clear that this is an experimental retrospective study. Then, the authors should avoid to refer to the term "micronutrients" since they only analyzed minerals.
Author Response
March 1, 2022
Dear editor:
Thank you for your encouraging letter concerning our manuscript entitled “Association between dietary micronutrients and retinopathy” by Yuan-Yuei Chen et al, for publication in Article of Nutrients.
We are extremely grateful to you and the reviewers for the constructive critique of our manuscript. We have responded to each of the comments of the referees on separate sheets and deeply appreciated your suggestions that have led to a significant improvement in this article. In response to your comments, we have revised the manuscript to enhance article readability. Several new sections of text and tables are added. We have also reedited the abstract, introduction, results, methods, and discussion sections. All the changes are labeled in red color. This manuscript has been edited by the American Journal Experts. Hope our revisions have fully addressed your concerns. Accordingly, we resubmit this article to Nutrients.
We look forward to your prompt reply.
Yours sincerely,
Wei-Liang Chen, M.D. Ph.D.
Division of Geriatric Medicine, Department of Family Medicine, Tri-Service General Hospital, National Defense Medical Center,
Number 325, Section 2, Chang-gong Rd, Nei-Hu District, 114, Taipei, Taiwan, ROC.
Tel: +886-2-87923311 ext. 16567
Fax: +886-2-87927057
E-mail: weiliang0508@gmail.com
Answer to Reviewer's comments
Thank you for your positive comments on this manuscript. The responses to the raised questions are below.
Reviewer 2
General comment:
- I’ve read with attention the paper of Chen and Chen that is potentially of interest. The background and aim of the study have been clearly defined. The methodology applied is overall correct, the results are reliable and adequately discussed. However, the authors should make the title more informative: it should be clear that this is an experimental retrospective study. Then, the authors should avoid to refer to the term "micronutrients" since they only analyzed minerals.
Response: Thank you for your thorough review and salient observations. We have revised the title based on your recommendation.
Association between dietary calcium and potassium and retinopathy: a cross-sectional retrospective study
Last, we are deeply honored by the time and effort you spent in reviewing this manuscript. In reviewing and revising our text, we are motivated to read more and thus learn more from your criticisms.
